# Numerical simulation and experimental verification of venturi tube hydraulic cavitation

**Zhanshuo Zhang**[1☉], **Sitong Guo**[1☉], **Xueying Ji**[1], **Linlin Cao**[1*], **Zhanshan Ma**[1‡], **Yunsheng Tian**[1], **Xiaolong Zhou**[1*], **Zhijie Huang**[2‡], **Xiaobo Liu**[1]

**1** Mechanical Engineering College, Beihua University, Jilin, Jilin Province, China, **2** Mechanical Engineering College, Guangxi University, Nanning, Guangxi Province, China

☉ These authors contributed equally to this work.
‡ ZM and ZH also contributed equally to this work.
* caolinlin0626@126.com (LC); xlzhou1987@163.com (XZ)

## Abstract

This study conducted a numerical simulation of hydraulic cavitation characteristics in a Venturi tube using FLUENT software. The Realizable k-ε turbulence model, Mixture multiphase flow model, and Singhal cavitation model were employed to investigate the effects of inlet pressure, outlet cone angle, and throat parameters (diameter and length) on cavitation performance. A critical inlet pressure threshold (~1.5 MPa) exists, beyond which the cavitation growth rate significantly decreases. Increasing the outlet cone angle weakens cavitation intensity due to reduced pressure recovery efficiency. Larger throat diameters enhance cavitation generation, whereas extended throat lengths suppress it by prolonging pressure recovery. Experimental validation demonstrated consistent trends between temperature variations, conductivity measurements, and simulation results, confirming the validity of the numerical methodology. These findings provide theoretical guidance for optimizing Venturi tube structures in industrial applications such as wastewater treatment and chemical reactors. The systematic analysis of parameter interactions offers practical insights for cavitation control and device performance enhancement.

## I. Introduction

Hydraulic cavitation is a hydrodynamic phenomenon characterized by the generation, development, and collapse of cavitation bubbles when liquid flows through a flow-restricting structure, where a sharp increase in flow velocity leads to a rapid pressure drop below the saturated vapor pressure of the liquid at the corresponding temperature [1–4]. During bubble collapse, localized high-temperature and high-pressure environments are instantaneously formed, generating microjets and shock waves that release substantial energy. By effectively harnessing this released energy, hydraulic cavitation has emerged as a widely adopted process intensification method

**Data availability statement:** All relevant data are within the manuscript and its Supporting Information files.

**Funding:** This work was supported by the Science and Technology Development Plan Project of Jilin Province, China (Grant No. YDZJ202401602ZYTS). This study did not receive any additional external funding.

**Competing interests:** The authors have declared that no competing interests exist.

in industrial production, with extensive applications in copper ore flotation, organic pollutant degradation in water, nanomaterial synthesis, and chemical reactions [5–7].

The Venturi tube is one of the most widely used hydraulic cavitation devices in both laboratory and industrial settings, where its structural design and operational conditions critically influence cavitation intensity. The rapid advancement of computational technology and the maturation of computational fluid dynamics (CFD) software have facilitated extensive numerical investigations into hydraulic cavitation phenomena within Venturi tubes by researchers globally. For instance, Zhang, Liu and Li [8–10] employed FLUENT-based simulations to systematically analyze the effects of structural parameters—such as contraction ratio and diffusion angle—on mass flux and flow symmetry. Their studies further quantified the impact of throat diameter on cavitation efficiency and validated the accuracy of the standard k-ε turbulence model combined with a hybrid cavitation model in predicting the hydraulic performance of Venturi injectors. Additionally, Zhang et al. [11] proposed a modified Zwart model, replacing the conventional cavitation number with the dimensionless parameter Pr, to elucidate the spatiotemporal relationship between cavitation clouds and vortex structures under cryogenic conditions. Concurrently, Shi, Pouffary and Chen [12–14] utilized CFD simulations to optimize structural parameters of Venturi-based cavitation devices, including throat geometry, throat length, convergence/divergence angles, and perimeter-to-area ratios, demonstrating significant enhancements in cavitation performance and providing practical design insights. Wang Zhiyong [15] further investigated the influence of operational parameters (e.g., pressure and temperature) on cavitation intensity, revealing that elevated inlet pressure amplifies cavitation effects under constant outlet pressure. Moreover, Gutiérrez-Montes, Zhao and Cai [16–18] explored the effects of structural parameters on cavitation stability, predicting critical pressure ratios and mass flow rates under cavitation conditions. In addition, Yuan, Junjie et al. and Gardenghi et al. have shown that, for conductivity-based measurement techniques, variations in liquid-phase conductivity associated with the final temperature difference can lead to a significant correlation between the measured average void fraction and both the final temperature difference and the liquid-phase conductivity when effective compensation is not applied [19–20]. However, a notable limitation persists in current studies [21–22]: most numerical models isolate the Venturi tube itself, neglecting the upstream and downstream straight pipe segments typically connected in practical applications. This simplification may introduce discrepancies between simulated results and real-world performance, emphasizing the necessity of incorporating full-scale piping configurations to rigorously evaluate their impact on cavitation flow fields.

In this study, numerical simulations were conducted using the Realizable k-ε turbulence model, Mixture multiphase flow model, and Singhal cavitation model to analyze cavitation flow within venturi tubes under specified operating conditions. With the tube outlet pressure fixed at 0.11 MPa (assumed condition), variations in inlet pressure and structural parameters were systematically examined to determine their effects on void fraction distribution. The obtained void fraction patterns provide valuable insights for optimizing the design of venturi devices and enhancing

cavitation-induced process intensification applications. This research contributes to a deeper understanding of flow field dynamics within venturi tubes and offers practical for engineering optimizations.

## II. Experiments and materials

### A. Experiment preparation

The throttling hole type microbubble generator processing equipment used in this study is the FDM 3D printer (P1S Combo, Shenzhen Tuo Zhu Technology Co., Ltd., Shenzhen, China). The materials utilized in the research include PVA (water-soluble printing consumables, Shenzhen Tuo Zhu Technology Co., Ltd.), ABS (printing consumables, Shenzhen Tuo Zhu Technology Co., Ltd.), printer bed adhesive (0.5 kg, Shenzhen Tuo Zhu Technology Co., Ltd.), sandpaper (400/1200/2000 grit, Beijing Dongxin Abrasive Tools Co., Ltd., Beijing, China), etc.

When preparing the throttling hole type microbubble generator, PLA material was primarily chosen, and the parts were printed using 3D printing technology, as shown in Fig 1. The manufacturing steps for the throttling hole type microbubble generator are as follows: design the three-dimensional model of the throttling hole type microbubble generator using SolidWorks software; import the model into Magics software to adjust the printing parameters; Calibrate the 3D printer; apply adhesive on the heated bed of the printer; Print the throttling hole type microbubble generator; Remove the part from the heated bed; Dissolve the printer supports in water; use sandpaper to polish the part surface; Place the part in an oven at 45°C for drying; Obtain a throttling hole type microbubble generator.

### B. Experiment Setup

A complete venturi tube cavitation performance test device is constructed using a combined venturi tube as the carrier, as shown in Fig 2. This device mainly consists of a venturi tube cavitation generator (PLA, self-made), a storage tank (140L), a water pump (JTP-4800 32W, Shanghai Oriental Pump Industry Group Co., Ltd., Shanghai, China), a globe valve (DN25, Shanghai Hugong Valve Factory (Group) Co., Ltd.), a high-pressure pump (Q(D)3–165/10–5, Shanghai Kechuang People's Co., Ltd., Shanghai, China), a turbine flowmeter (LWGY-G, Shanghai Huiyi Co., Ltd., Shanghai, China), a conductivity meter (AR8011, Shanghai Sima Technology Co., Ltd., Shanghai, China), a temperature gun (IM-9001, Shenzhen

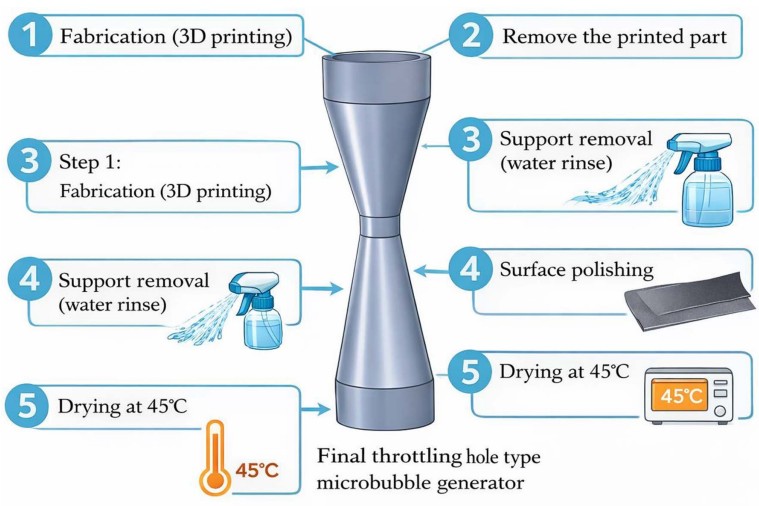

**Fig 1. The fabrication process of throttling hole type microbubble generator.**

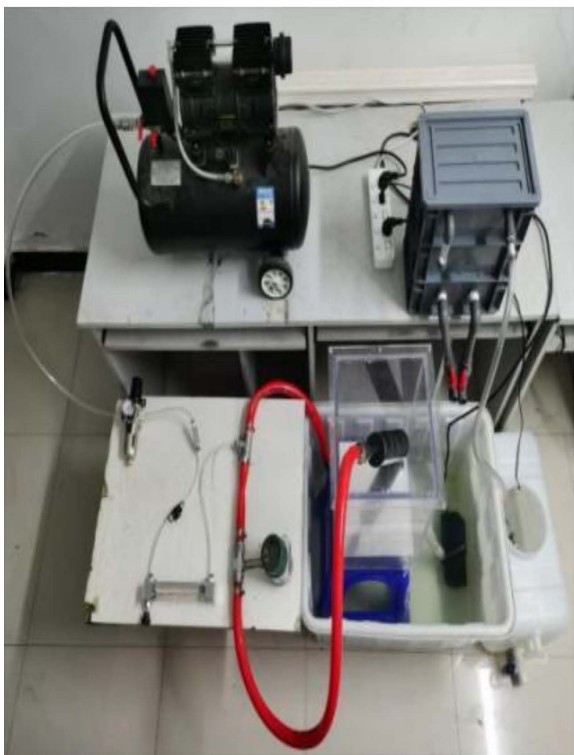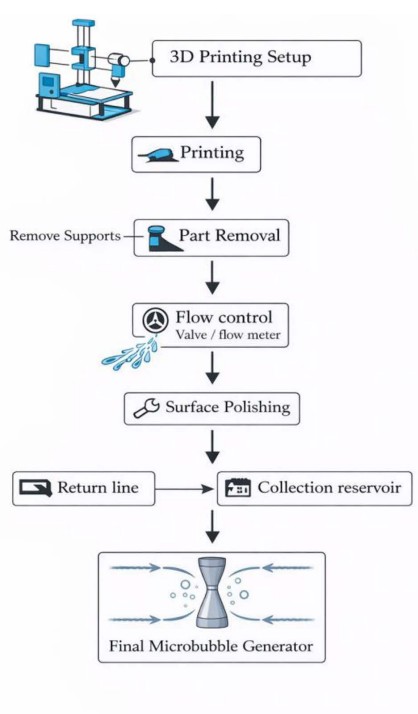

**Fig 2. Schematic diagram of the experimental setup and flow procedure used in the Venturi tube cavitation experiments.**

Zhuochen Technology Co., Ltd., Shenzhen, China), etc. The water pump, globe valve, turbine flow meter, and water tank form the liquid supply unit, while the conductivity meter and thermometer constitutes the detection unit.

Hydrodynamic cavitation experiments were conducted using the experimental apparatus shown in Fig 2. After 60 minutes of cavitation, samples were taken. Pure water was used to set up a blank control group, and the physical properties such as temperature and conductivity of the samples were measured. The energy of cavitation can break the O-H bonds between water molecules, decomposing water into hydroxyl and hydrogen radicals. Additionally, some other free radicals and even a certain amount of new charged particles are generated in the liquid, thereby enhancing the electrical conductivity of the water. To verify the accuracy of the numerical simulation results, verification experiments of venturi tube cavitation were carried out under different working pressures and structural parameters. The cavitation effect of the venturi tube was characterized by conductivity and water temperature. During the unit cavitation time, the higher the conductivity or temperature rise of the water sample, the better the cavitation effect.

## III. Numerical section

### A. Turbulence model

**(1) Turbulence modelling.** In the numerical simulation of cavitation in Venturi tubes, the turbulence model not only influences computational accuracy and the prediction of reflected flows but also governs the motion of vortex clusters, thereby underscoring the critical importance of turbulent model selection. Currently, mainstream turbulence calculation models include Direct Numerical Simulation (DNS), Large Eddy Simulation (LES), and Reynolds-Averaged Navier-Stokes (RANS) approaches. Based on previous numerical simulation experience and a comprehensive review of relevant literature, the Reynolds-Averaged Navier-Stokes (RANS) method has been identified as the most appropriate choice.

Among RANS models, two-equation models demonstrate superior universality. To ensure compliance with fundamental physical laws governing turbulent flows, the present study adopts the Realizable k-ε model. The transport equations for turbulent kinetic energy and dissipation rate are formulated as follows:

$$\frac{\partial (\rho k)}{\partial t} + \frac{\partial (\rho k \mu_i)}{\partial x_i} = \frac{\partial}{\partial x_j}\left[\left(\mu + \frac{\mu_t}{\sigma_k}\right)\frac{\partial k}{\partial x_j}\right] + G_K + G_b + \rho\varepsilon + Y_M \tag{1}$$

$$\frac{\partial (\rho\varepsilon)}{\partial t} + \frac{\partial (\rho\varepsilon\mu_i)}{\partial x_i} = \frac{\partial}{\partial x_j}\left[\left(\mu + \frac{\mu_t}{\sigma_\varepsilon}\right)\frac{\partial \varepsilon}{\partial x_j}\right] + \rho C_1 E\varepsilon - \rho C_2\frac{\varepsilon^2}{k + \sqrt{\upsilon\varepsilon}} + C_{1\varepsilon}\frac{\varepsilon}{k}C_{3\varepsilon}G_b \tag{2}$$

In the formula, ρ represents the liquid density, μ represents the liquid dynamic viscosity, μt represents the turbulent eddy viscosity, Gk represents the turbulent kinetic energy production due to mean velocity gradients, Gb represents the turbulent kinetic energy production due to buoyancy, YM represents the contribution of compressible turbulence to dissipation rate, C1ε, C3ε, C2ε represents the model constants, (default values in FLUENT: C1ε = 1.44, C3ε = 0.99, C2ε = 1.9), σk, σε represents the turbulent Prandtl numbers for k and ε equations (σk = 1.0, σε = 1.5).

**(2) Multiphase flow model.** During the cavitation process in Venturi tubes, the gas-liquid two-phase flow exhibits random distribution of phases within the liquid medium, accompanied by continuous dynamic interactions. The commonly employed computational models for cavitation two-phase flow include the Eulerian Model, Volume of Fluid (VOF) Model, and Mixture Model. Based on numerical simulation experience and literature review, Venturi tube cavitation has been identified as a process involving mass transfer between gas and liquid phases, with multiple factors influencing cavitation outcomes. Therefore, the mixture model was selected for this analysis. The governing equations consist of the continuity equation and momentum equation, expressed as follows:

$$\frac{\partial \rho_m}{\partial t} + \frac{\partial (\rho_m u_j)}{\partial x_j} = 0 \tag{3}$$

Further derivation yields:

$$\frac{\partial}{\partial x_i}(\rho_m u_i u_j) = \frac{\partial}{\partial x_i}\left[\mu_f\left(\frac{\partial u_i}{\partial x_j} + \frac{\partial u_j}{\partial x_i}\right) - \frac{2}{3}\frac{\partial}{\partial x_i}\left(\mu\frac{\partial u_j}{\partial x_j}\right) + \sum_{s=1}^{2}\alpha_s\rho_s u_{si}^r u_{sj}^r\right] - \frac{\partial p}{\partial x_i} + \rho_m g_j + F_j \tag{4}$$

In the formula,The subscripts i and j denote the directions of the coordinates. ρm, ρ and u respectively denote the density, pressure, and velocity of the mixing phase.; Fj is the volume force; ursj, ursi is the relative slip velocity between the two phases; μf is the effective viscosity between the mixture phases; ρs is the density of the s phase; αε is the volume ratio of the s phase; μ is the laminar viscosity coefficient between the mixture phases.

**(3) Cavitation model.** To comprehensively account for the complex interactions arising from liquid surface tension, non-condensable gas concentration, and liquid vaporization pressure, the Singhal cavitation model based on the Full Cavitation Model framework was employed to solve the mass transfer process in liquid-vapor two-phase flows. The transport equation governing the vapor mass fraction can be expressed as:

$$\frac{\partial}{\partial t}(f_v\rho) + \nabla.\left(f_v\rho\overrightarrow{V_v}\right) = \nabla.(\Gamma\nabla f_v) + R_e - R_c \tag{5}$$

If the pressure is less than the saturated vapor pressure, then evaporation occurs:

 

$$R_e = F_{vap} \frac{\max\left(1.0, \sqrt{k}\right)(1 - f_v - f_g)}{\sigma} \rho_l \rho_v \sqrt{\frac{2}{3} \frac{(P_v - P)}{\rho_l}} \qquad (6)$$

If the pressure is greater than the saturated vapor pressure, then condensation occurs.

In the formula:

$$R_c = F_{cond} \frac{\max\left(1.0, \sqrt{k}\right) f_v}{\sigma} \rho_l \rho_v \sqrt{\frac{2}{3} \frac{(P - P_v)}{\rho_l}} \qquad (7)$$

In the formula, fv is the mass fraction of steam; fg is the mass fraction of non-condensable gas; Γ is the diffusion coefficient; Re is the evaporation rate; Rc is the condensation rate; Fvap is the evaporation coefficient, Fcond is the evaporation coefficient, the values of Fvap and Fcond are Fvap = 0.02 and Fcond = 0.01, respectively.

## B. Geometric model

The geometric model of the Venturi tube employed in this study is illustrated in Fig 3. The numerical simulations were conducted with a fixed outlet pressure of 0.11 MPa and varying inlet pressures ranging from 0.2 MPa to 2.0 MPa (specifically 0.2, 0.3, 0.4, 0.5, 0.7, 0.9, 1.0, 1.5, and 2.0 MPa). This pressure range was selected to encompass typical flow measurement applications (e.g., chemical pipelines and water treatment systems) while simultaneously investigating cavitation characteristics under high-pressure conditions. The divergence angles at the outlet section were systematically varied across nine configurations (6°, 8°, 10°, 12°, 15°, 20°, 25°, 30°, 35°, and 40°). Smaller divergence angles (6°-15°) facilitate gradual flow expansion in the outlet section, promoting flow stability recovery while minimizing flow separation and energy dissipation – characteristics particularly advantageous for high-precision flow measurement applications. Conversely,

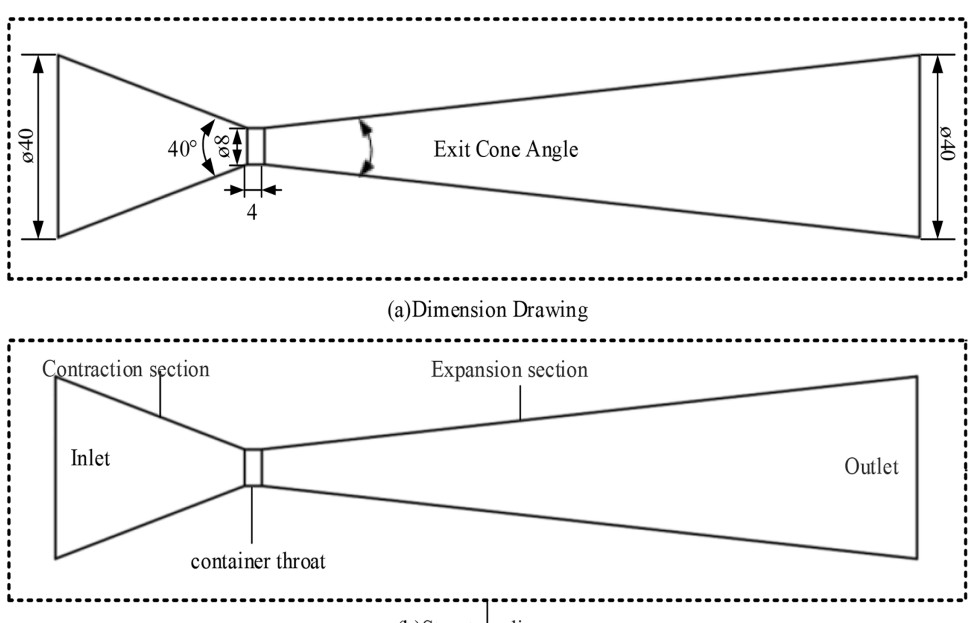

**Fig 3. Schematic of the venturi tube model.**

larger divergence angles (20°-40°) may induce flow separation and vortex formation but offer the practical benefit of reduced pipe length, making them suitable for space-constrained engineering applications. Comparative analysis of flow field characteristics across different divergence angles enables optimization of pressure recovery efficiency versus structural compactness.

To enhance numerical simulation accuracy, three-dimensional modeling was implemented rather than two-dimensional approximations. This approach enables more realistic reproduction of three-dimensional flow effects, particularly critical in the throat contraction region and outlet diffuser section. Three-dimensional modeling accounts for phenomena such as sidewall boundary layer development and asymmetric vortex structures that might be overlooked in two-dimensional simulations, particularly regarding the influence of radial velocity gradients on cavitation distribution patterns. The principal simulation parameters are summarized in Table 1.

## C. Mesh generation

A three-dimensional computational domain was established based on the geometric model of the experimental Venturi tube, with the inlet cross-section serving as the reference origin (zero point). The computational grid was constructed using O-type hexahedral structured elements. To address the intensified cavitation phenomena observed in the throat region, a partitioned meshing approach was implemented. This strategy combined local grid refinement in the throat area with graded mesh transitions in adjacent zones, achieving an optimal balance between computational accuracy and resource efficiency. Specifically, the throat region underwent densification through mesh compression techniques, ensuring sufficient resolution for capturing complex cavitation dynamics. The final meshing configuration is presented in Fig 4, demonstrating the spatial distribution of grid elements and the localized refinement scheme.

## D. Mesh independence verification

The grid resolution exerts a significant influence on the accuracy of numerical simulations. Meaningful simulation results can only be obtained when the grid density reaches a critical threshold where key computational parameters demonstrate minimal variation with further mesh refinement. To verify grid independence, four mesh configurations containing 3, 4, 5,

**Table 1. Venturi tube structural parameters.**

| Parameter | Value |
|---|---|
| Inlet/outlet diameter (mm) | 40 |
| Inlet convergent angle (°) | 40 |
| Outlet divergent angle (°) | 6/8/10/12/15/20/25/30/35/40 |
| Throat diameter (mm) | 8 |
| Throat length (mm) | 4 |

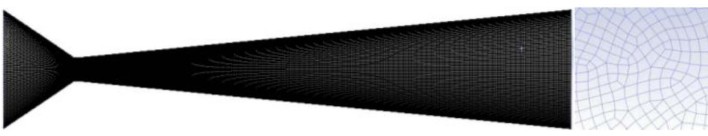

## (a)Overall diagram          (b) Detail diagram

**Fig 4. Mesh Generation Diagram. (a)** Overall diagram **(b)** Detail diagram.

and 6 million elements were established. The pressure distributions along the central axis were compared across different mesh densities to assess grid sensitivity, as shown in Fig 5. The analysis revealed that pressure distribution along the axis stabilized when the grid count reached 5 million elements, with negligible variations observed upon further mesh refinement.

Notably, a distinct pressure deviation was identified at the 160 mm (0.16 m) cross-section. To comprehensively validate grid independence, the turbulent dissipation rate at this specific cross-section was additionally analyzed. Table 2 presents the calculated average turbulent dissipation rates and corresponding relative deviations for each mesh configuration. The relative deviation decreased from 0.83% when refining from 3 to 4 million elements, to 0.38% when increasing from 4 to 5 million elements. Comparative analysis of both pressure distribution along the central axis and turbulent dissipation rates at the critical cross-section confirmed that the 5-million-element mesh configuration adequately satisfies grid independence requirements.

### E. Boundary conditions and solution algorithm

Since the flow within the Venturi tube involves the gas-liquid two-phase flow with irregular mixture of gas and liquid, the Mixture multiphase flow model was employed to ensure computational stability and efficiency. Considering the flow inside the Venturi tube was fully turbulent, the standard k-ε model was selected for turbulence simulation due to its suitability for such turbulent flow regimes. The PISO algorithm was utilized to couple pressure and velocity fields, which operates through a three-step process: one prediction step followed by two correction steps. Each step offers superior

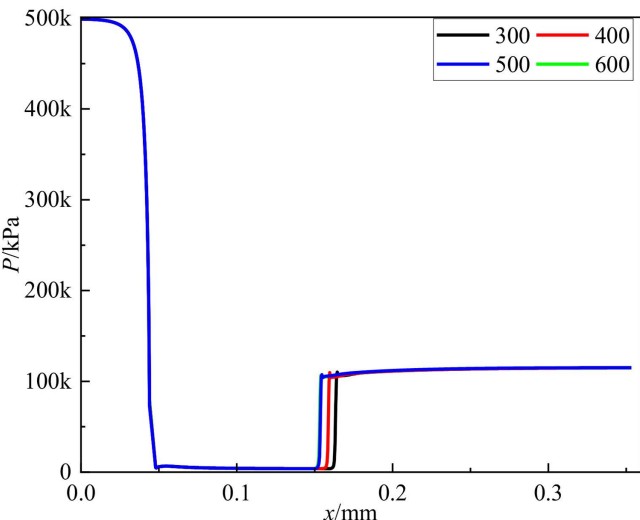

**Fig 5. Grid independence verification.**

**Table 2. Grid independence verification results.**

| Serial number | Grid number/10,000 | Average turbulence dissipation/$m^2 \cdot s^{-3}$ | Relative deviation/% |
|---|---|---|---|
| 1 | 300 | 110.98 | – |
| 2 | 400 | 105.73 | 4.73 |
| 3 | 500 | 104.85 | 0.83 |
| 4 | 600 | 104.45 | 0.38 |

computational accuracy and convergence properties compared to other algorithms. The pressure term was discretized using the PRESTO scheme, while other terms employed the quick scheme. The turbulence intensity was set at 3%, with an initial gas volume fraction of 0 and a cavitation critical pressure of 3540 Pa. Standard wall functions were applied for near-wall calculations.

## IV. Numerical analysis

When water flows through the Venturi tube and passes through the throat, the cross-sectional area of the pipeline becomes smaller, the water flow velocity increases, and the pressure decreases. When the pressure drops below the saturation vapor pressure of the ambient temperature, water undergoes a gas phase transformation, leading to the explosive growth, expansion, contraction, and collapse of micro-gas nuclei in the water, a process known as hydrodynamic cavitation. The generation of cavitation is accompanied by a series of high-pressure and high-speed jet phenomena. Reasonable utilization of cavitation can improve production efficiency and save energy.

### A. Influence of inlet pressure on cavitation effect

The cloud map of the gas holdup variation at different inlet pressures is shown in Fig 6. Among them, Fig 6(a) represents the cloud map of the cavitation area distribution under different inlet pressures; Fig 6(b) shows the relationship between different inlet pressures and gas content. According to the data changes reflected in the Figures, it can be concluded that the average gas holdup rate in the Venturi tube gradually increases with the increase in inlet pressure, and the cavitation intensity shows a trend of gradual increase. When the inlet pressure is 0.2 MPa-0.7 MPa, the growth is slow; when the inlet pressure is 0.9 MPa-2.0 MPa, the increment in gas holdup rate becomes smaller. The influence of inlet pressure of the cavitation effect of the Venturi tube is not infinitely positively correlated, but there is a limit value. The cavitation area in the Venturi tube increases with the increase in inlet pressure, showing a positive correlation. Table 3 demonstrated the average gas fraction and maximum gas fraction of the Venturi tube under different inlet pressures, under the conditions of a 40° inlet cone angle and a 6° outlet cone angle.

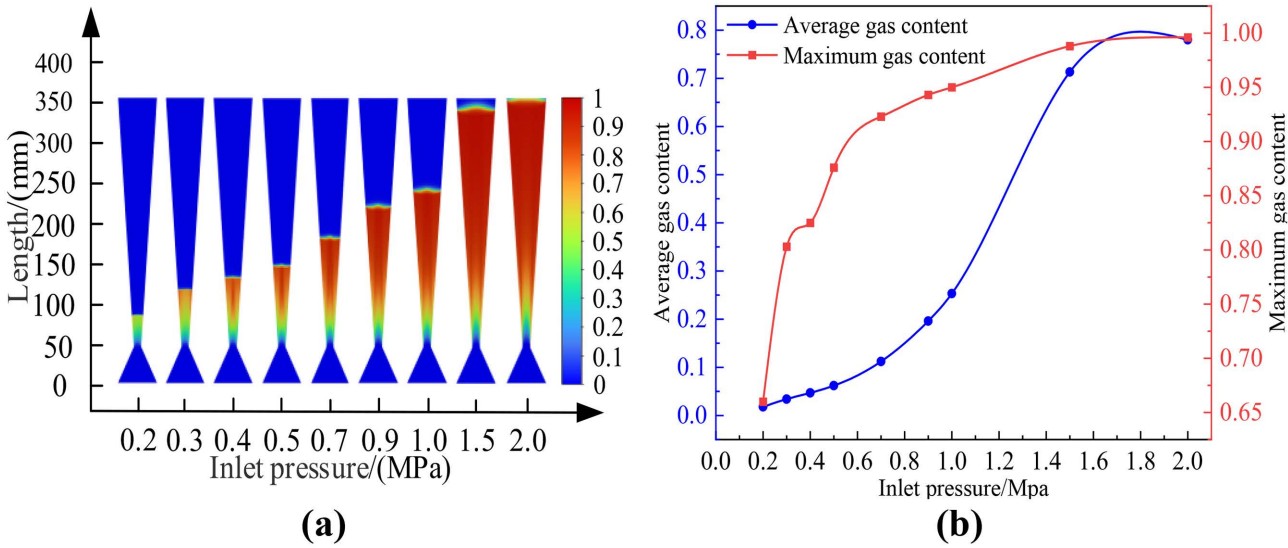

**Fig 6. Cloud maps of gas holdup variations at different inlet pressures.**

**Table 3. Inlet Pressure vs. Gas Content. (a-b).**

| Venturi tube | Inlet cone angle: 40°, Outlet cone angle: 6° | | | | | | | | |
|---|---|---|---|---|---|---|---|---|---|
| Inlet pressure/MPa | 0.2 | 0.3 | 0.4 | 0.5 | 0.7 | 0.9 | 1.0 | 1.5 | 2.0 |
| Average gas content | 0.018 | 0.034 | 0.047 | 0.062 | 0.112 | 0.196 | 0.253 | 0.713 | 0.780 |
| Maximum gas content | 0.660 | 0.803 | 0.825 | 0.876 | 0.923 | 0.943 | 0.950 | 0.988 | 0.996 |

## B. Influence of outlet cone angle on cavitation effect

**(1) Constant length of venturi tube.** Using the standard k-ε turbulence model and the Mixture multiphase flow model, numerical simulations were conducted on a Venturi tube with inlet and outlet diameters of 40 mm, throat diameter of 8 mm, inlet cone angle of 40 degrees, and inlet and outlet pressures of 0.5 MPa and 0.115 MPa respectively. The influence of different outlet cone angles on the cavitation flow field was studied. In the simulation, the cavitation effect was measured by the change in gas content. The obtained results are presented in Fig 7. In this figure, (a) shows the distribution cloud map of the cavitation area for different outlet cone angles; (b) shows the relationship between different outlet cone angles and gas content. Table 4 showed the average and maximum gas holdup of the Venturi tube at different outlet cone angles, with the specific conditions of an inlet cone angle of 40° and an inlet pressure of 0.5 MPa.

As shown in Fig 7(b), the average gas holdup rate in the Venturi tube gradually decreases with the increase in outlet cone angle, and the decrease is slow and the value is small when the outlet cone angle is 25°-40°, at which time there is no cavitation in the Venturi tube. The maximum gas holdup rate in the Venturi tube first increases and then decreases

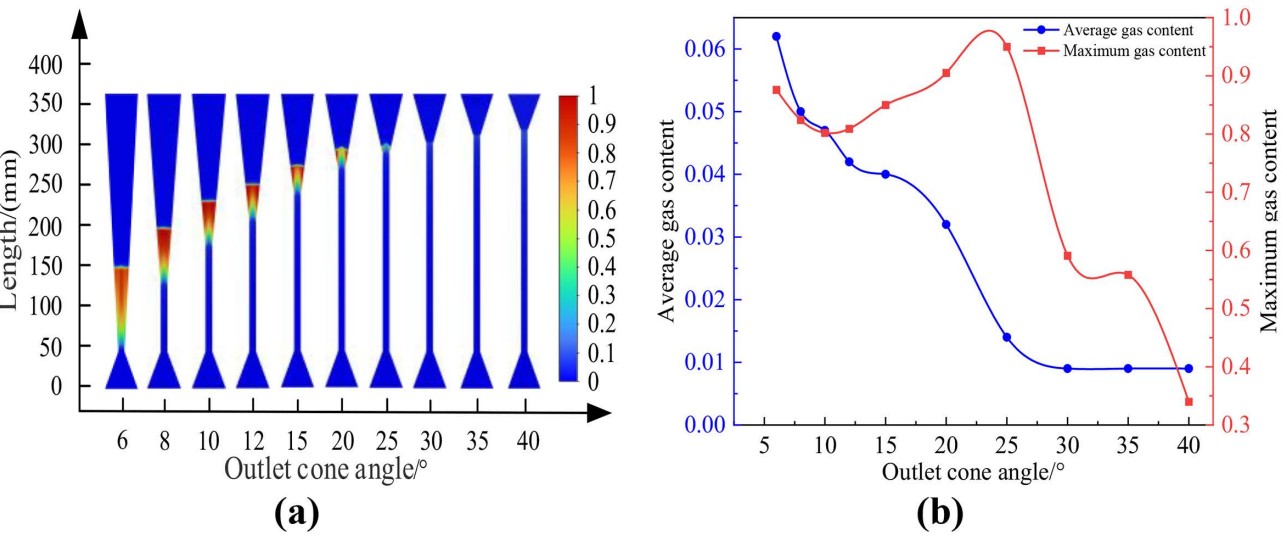

**(a)** **(b)**

**Fig 7. Cloud images with variations in exit cone angle and gas content rate.** (a-b).

**Table 4. Relationship between outlet cone angle and gas content.**

| Venturi tube | Inlet cone angle: 40°, Inlet pressure: 0.5 MPa | | | | | | | | | |
|---|---|---|---|---|---|---|---|---|---|---|
| Outlet cone angle | 6° | 8° | 10° | 12° | 15° | 20° | 25° | 30° | 35° | 40° |
| Average gas content | 0.062 | 0.050 | 0.047 | 0.042 | 0.040 | 0.032 | 0.014 | 0.009 | 0.009 | 0.009 |
| Maximum gas content | 0.876 | 0.824 | 0.802 | 0.809 | 0.850 | 0.905 | 0.950 | 0.591 | 0.558 | 0.340 |

with the increase in outlet cone angle, and slowly increases when the outlet cone angle is 10°-25°, and rapidly decreases when the outlet cone angle is 25°-40°.

According to the simulation results in Fig 8, it can be concluded that when the inlet and outlet pressures are constant and the pipe length unchanged, the larger the outlet cone angle, the smaller the maximum gas holdup rate along the central axis of the Venturi tube and the axial range from the generation to disappearance of bubbles. Because when other conditions are constant, the larger the outlet cone angle, the longer the throat length, the shorter the pressure recovery section, leading to an increase in pressure and a decrease in velocity in the pressure recovery section, an increase in the cavitation number, and therefore a decrease in cavitation intensity. In summary, it is concluded that the cavitation intensity gradually decreases with the increase in outlet cone angle.

**(2) Variable length of venturi tube.** Using the standard k-ε turbulence model and Mixture multiphase flow model, the cavitation flow field in the Venturi tube with different pipe lengths and outlet cone angles is numerically simulated under the conditions of inlet and outlet diameters of 40 mm, throat diameter of 8 mm, throat length of 4 mm, inlet cone angle of 40 degrees, inlet and outlet pressures of 0.5 MPa and 0.115 MPa.The obtained results are presented in Fig 9. Specifically, Fig 9(a) shows the distribution cloud map of cavitation regions for different outlet cone angles; Fig 9(b) depicts the relationship between different outlet cone angles and the gas content rate. Table 5 demonstrated the average gas fraction and maximum gas fraction of the Venturi tube with varying outlet cone angles after changing the tube length, under the conditions of a 40° inlet cone angle and 0.5 MPa inlet pressure.

According to Fig 9(b), the average gas holdup rate in the Venturi tube first increases and then decreases with the increase in outlet cone angle, and slowly increases when the outlet cone angle is 6°-12°, and rapidly decreases when the outlet cone angle is 12°-40°, and the average gas holdup rate is small when the outlet cone angle is 30°-40°, at which time there is no cavitation in the Venturi tube. The maximum gas holdup rate in the Venturi tube first increases and then decreases with the increase in outlet cone angle, and slowly increases when the outlet cone angle is 6°-30°, and rapidly decreases when the outlet cone angle is 30°-40°.

According to the simulation results in Fig 10, it can be concluded that when the inlet and outlet pressures are constant and the pipe length changes, the cavitation intensity in the Venturi tube gradually decreases with the increase in outlet

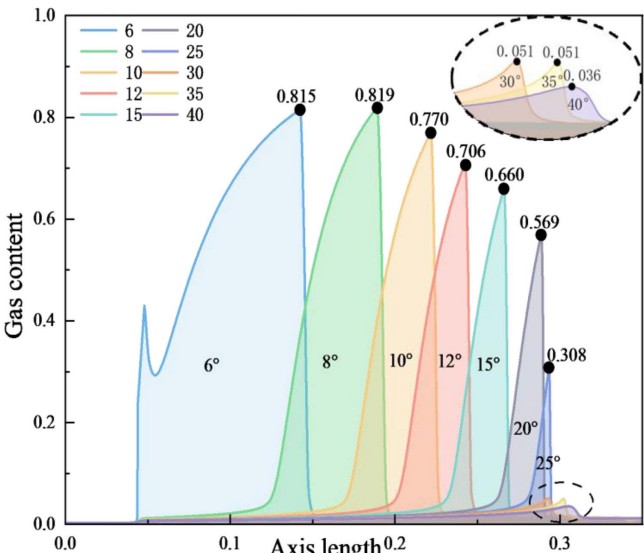

**Fig 8. Relationship between outlet cone angle and gas content distribution on central axis.**

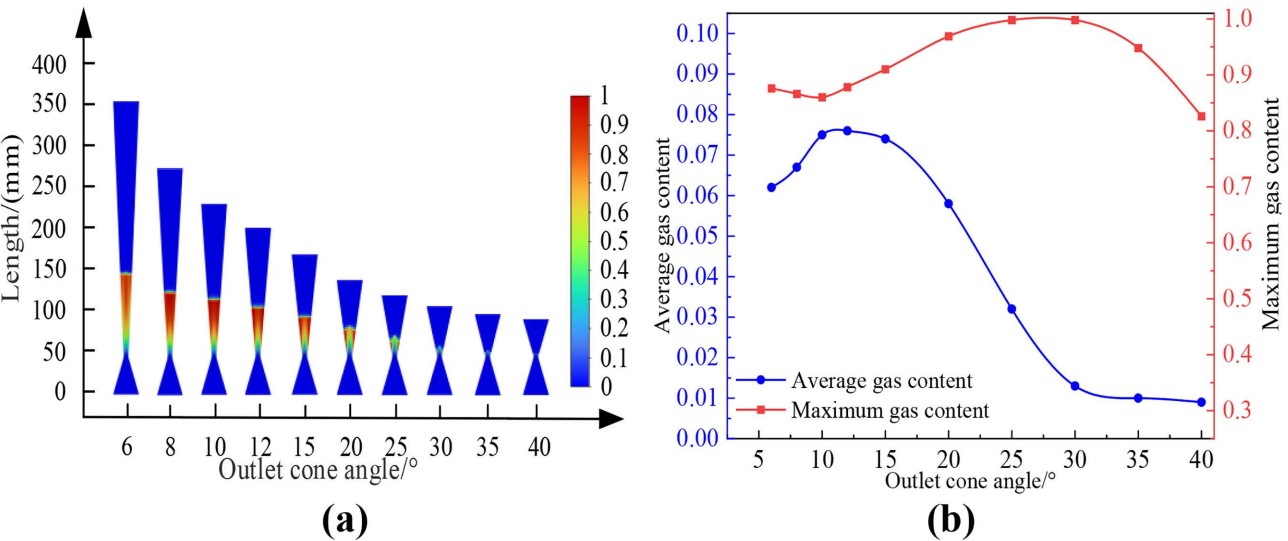

**Fig 9. Cloud images with variations in exit cone angle and gas content rate.** (a-b).

**Table 5. Relationship between outlet cone angle and gas content.**

| Venturi tube | Inlet cone angle: 40°, Outlet cone angle: 0.5 MPa | | | | | | | | | |
|---|---|---|---|---|---|---|---|---|---|---|
| Outlet cone angle | 6° | 8° | 10° | 12° | 15° | 20° | 25° | 30° | 35° | 40° |
| Average gas content | 0.062 | 0.067 | 0.075 | 0.076 | 0.074 | 0.058 | 0.032 | 0.013 | 0.010 | 0.009 |
| Maximum gas content | 0.876 | 0.866 | 0.860 | 0.878 | 0.910 | 0.969 | 0.998 | 0.998 | 0.948 | 0.826 |

cone angle, and the maximum gas holdup rate and the axial range from the generation to disappearance of bubbles decrease. Because when other conditions are constant, the smaller the outlet cone angle, the longer the diffusion section flow field of the Venturi tube, leading to more complete cavitation and stronger cavitation intensity.

## C. Influence of throat parameters on cavitation effect

**(1) Influence of throat diameter – constant length of venturi tube.** Using the standard k-ε turbulence model and Mixture multiphase flow model, the cavitation flow field in the Venturi tube with different throat diameters and cone angles is numerically simulated under the conditions of inlet and outlet diameters of 40 mm, inlet cone angle of 40 degrees, outlet cone angle of 20 degrees, inlet and outlet pressures of 0.5 MPa and 0.115 MPa. The obtained results are presented in Fig 11. Among them, Figure 11 (a) shows the cloud map of the distribution of cavitation areas with different throat diameters; Figure 11 (b) shows the relationship between different throat diameters and gas holdup. Table 6 demonstrated the average gas fraction and maximum gas fraction of the Venturi tube with different throat diameters, under the conditions of a 40° inlet cone angle, a 40° outlet cone angle, and 0.5 MPa inlet pressure.

According to Fig 11(b), the average gas holdup rate in the Venturi tube gradually increases with the increase in throat diameter, and reaches the maximum value of 0.139 when the throat diameter is 12 mm. The maximum gas holdup rate in the Venturi tube remains basically unchanged with the increase in throat diameter, and the maximum gas holdup rate remains around 0.970.

In summary, it is concluded that when the length of the Venturi tube is constant, the increase in throat diameter has a positive impact on the average gas holdup rate in the Venturi tube, but has no impact on the maximum gas holdup rate, and the overall cavitation intensity gradually increases.

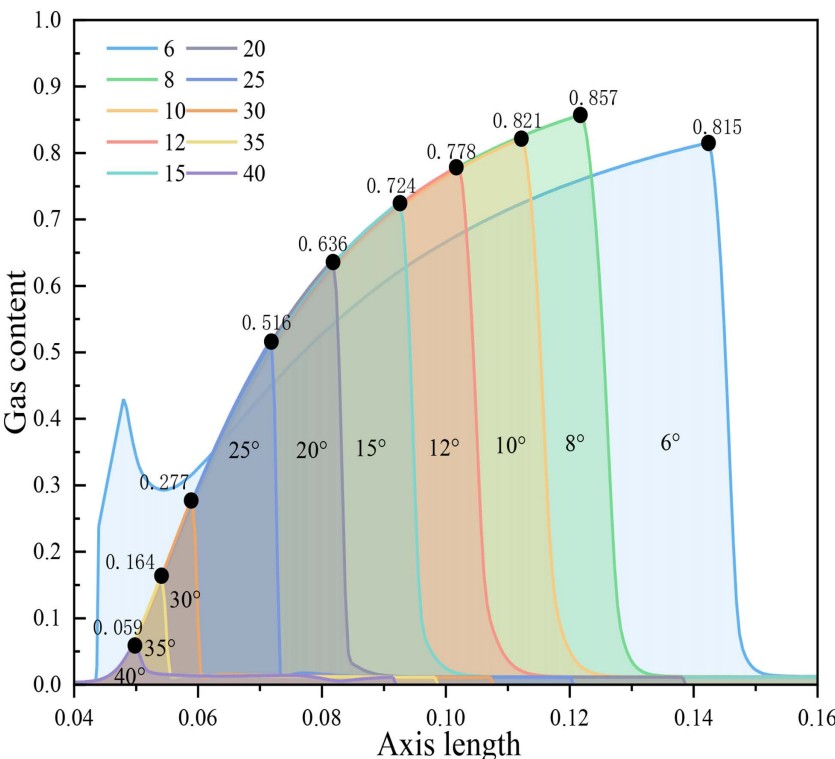

**Fig 10. Relationship between outlet cone angle and gas content distribution on central axis.**

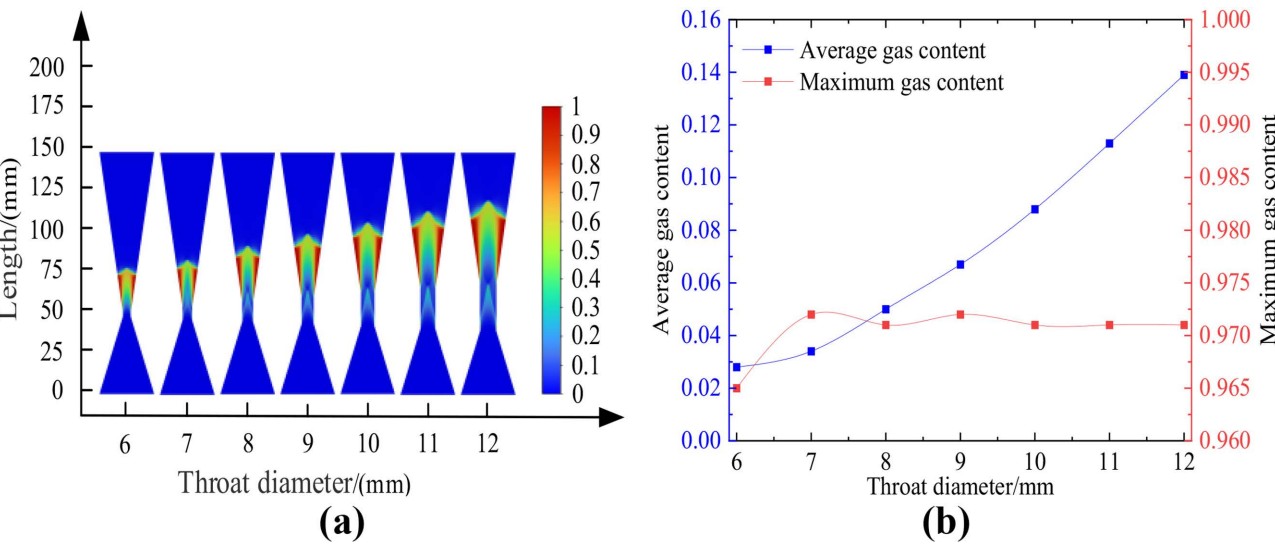

**Fig 11. Cloud images of changes in different throat diameters and gas content.** (a-b).

**Table 6. Relationship between throat diameter and gas content.**

| Venturi tube | Inlet cone angle: 40°, Outlet cone angle: 40°, Inlet pressure: 0.5 MPa | | | | | | |
|---|---|---|---|---|---|---|---|
| Throat diameter/mm | 6 | 7 | 8 | 9 | 10 | 11 | 12 |
| Average gas content | 0.028 | 0.034 | 0.050 | 0.067 | 0.088 | 0.113 | 0.139 |
| Maximum gas content | 0.965 | 0.972 | 0.971 | 0.972 | 0.971 | 0.971 | 0.971 |

**(2) Influence of throat diameter – variable length of venturi tube.** Using the standard k-ε turbulence model and Mixture multiphase flow model, the cavitation flow field in the Venturi tube with different throat diameters and cone angles is numerically simulated under the conditions of inlet and outlet diameters of 40 mm, inlet cone angle of 40 degrees, outlet cone angle of 20 degrees, throat length of 4 mm, inlet and outlet pressures of 0.5 MPa and 0.115 MPa.The obtained results are presented in Figure 12. Among them, Fig 12(a) shows the cloud map of the distribution of cavitation areas with different throat diameters; Fig 12(b) shows the relationship between different throat diameters and gas holdup. Table 7 demonstrated the average gas fraction and maximum gas fraction of the Venturi tube with different throat diameters after the tube length was changed, under the conditions of a 40° inlet cone angle, a 40° outlet cone angle, and 0.5 MPa inlet pressure.

According to Fig 12(b), the average gas holdup rate in the Venturi tube gradually increases with the increase in throat diameter, and reaches the maximum value of 0.151 when the throat diameter is 12 mm. The maximum gas holdup rate in the Venturi tube gradually increases with the increase in throat diameter, and reaches the peak value of 0.994 when the throat diameter is 12 mm.

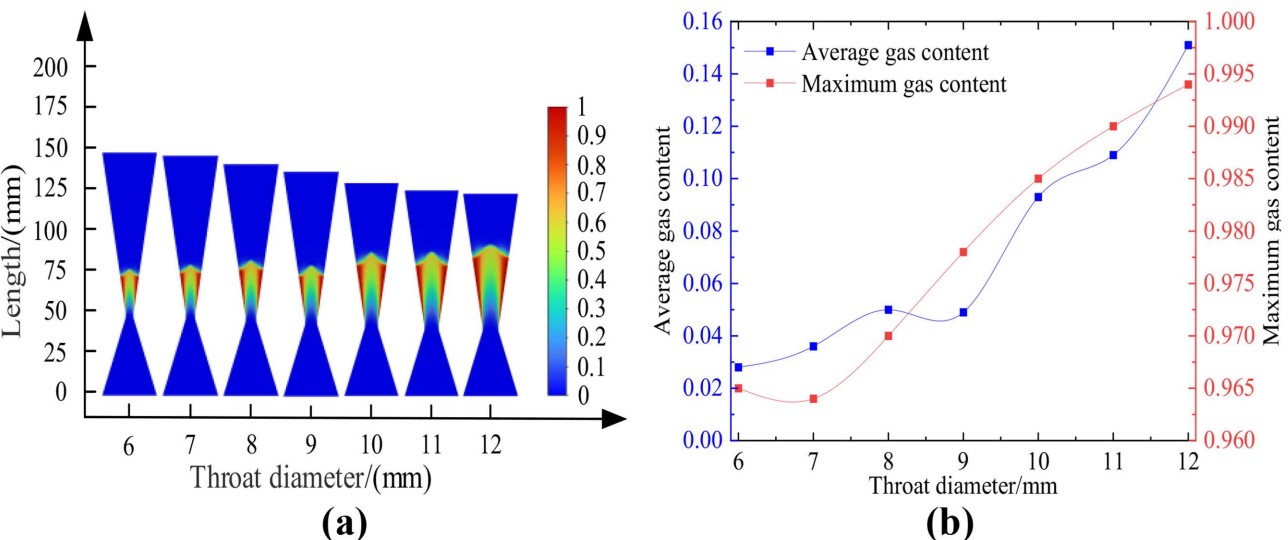

**Fig 12. Cloud images of changes in different throat diameters and gas content.** (a-b).

**Table 7. Relationship between throat diameter and gas content.**

| Venturi tube | Inlet cone angle: 40°, Outlet cone angle: 40°, Inlet pressure: 0.5MPa | | | | | | |
|---|---|---|---|---|---|---|---|
| Throat diameter/mm | 6 | 7 | 8 | 9 | 10 | 11 | 12 |
| Average gas content | 0.028 | 0.036 | 0.050 | 0.049 | 0.094 | 0.109 | 0.151 |
| Maximum gas content | 0.965 | 0.964 | 0.970 | 0.978 | 0.985 | 0.990 | 0.994 |

In summary, it is concluded that when the length of the Venturi tube changes, the increase in throat diameter has a positive impact on the average and maximum gas holdup rates in the Venturi tube, and the overall cavitation intensity gradually increases.

**(3) Influence of throat length.** Using the standard k-ε turbulence model and Mixture multiphase flow model, the cavitation flow field in the Venturi tube with different throat lengths and cone angles is numerically simulated under the conditions of inlet and outlet diameters of 40 mm, inlet cone angle of 40 degrees, outlet cone angle of 20 degrees, throat diameter of 10 mm, inlet and outlet pressures of 0.5 MPa and 0.115 MPa. The obtained results are presented in Figure 13. Among them, Fig 13(a) shows the cloud map of the cavitation area distribution with different throat lengths; Fig 13(b) shows the relationship between different laryngeal lengths and gas holdup. Table 8 demonstrated the average gas fraction and maximum gas fraction of the Venturi tube with different throat lengths, under the conditions of a 40° inlet cone angle, a 40° outlet cone angle, and 0.5 MPa inlet pressure.

According to Fig 13(b), the average gas holdup rate in the Venturi tube remains basically unchanged with the increase in throat length, and rapidly increases when the throat length is 0–4 mm, and there are small fluctuations but remains basically unchanged when the throat length is 4–40 mm. When the throat length is 0 mm, the average gas holdup rate is small because there is no throat, which makes the turbulence in the Venturi tube more intense, resulting in poor overall cavitation effect in the Venturi tube. The maximum gas holdup rate in the Venturi tube gradually decreases with the increase in throat length, and slowly decreases when the throat length is 0–8 mm, remains basically unchanged when the throat

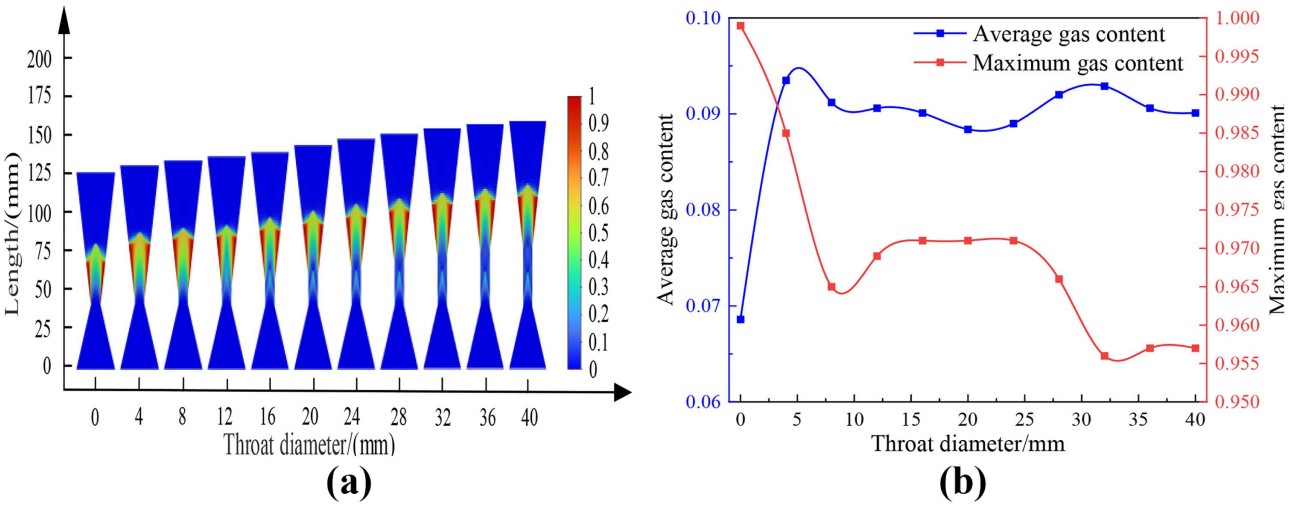

**(a)** **(b)**

**Fig 13. Cloud images of changes in different throat lengths and gas content rate.** (a-b).

**Table 8. Relationship between throat length and gas content.**

| Venturi tube | Inlet cone angle: 40°, Outlet cone angle: 40°, Inlet pressure: 0.5 MPa | | | | | |
|---|---|---|---|---|---|---|
| Throat diameter/mm | 0 | 4 | 8 | 12 | 16 | 20 |
| Average gas content | 0.069 | 0.094 | 0.091 | 0.091 | 0.090 | 0.088 |
| Maximum gas content | 0.999 | 0.985 | 0.965 | 0.969 | 0.971 | 0.971 |
| Throat diameter/mm | 24 | 28 | 32 | 36 | 40 | |
| Average gas content | 0.089 | 0.092 | 0.093 | 0.091 | 0.090 | |
| Maximum gas content | 0.971 | 0.966 | 0.956 | 0.957 | 0.957 | |

length is 12–24 mm, slowly decreases when the throat length is 24–32 mm, and remains basically unchanged when the throat length is 32–40 mm.

In summary, it is concluded that with the increase in throat length, the average gas holdup rate in the Venturi tube is hardly affected, the maximum gas holdup rate gradually decreases, and the overall cavitation intensity gradually decreases.

## V. Discussion of experimental results

### A. Experimental results of hydrodynamic cavitation

The experiments respectively explored the influences of inlet pressure, outlet cone Angle and throat diameter parameters on the cavitation effect of Venturi tubes. Fig 14 shows the changing trends of conductivity and temperature difference during the experiment. It can be seen from Figure 14 (a) that as the inlet pressure gradually increases, both the electrical conductivity and the temperature difference show a gradually increasing trend. However, the trend of temperature difference changes exponentially and is significantly different from the trends of average gas content rate and electrical conductivity. By consulting relevant literature, it was found that as the head of the high-pressure water pump increases, the heat generated by the pump also rises exponentially, which has a significant impact on the experimental results. To eliminate the interference of heat generated by other experimental equipment during operation on the experimental results, the experimental process of this experiment was redesigned. The specific method is to remove the Venturi tube cavitation unit and conduct experiments to obtain new experimental data. Subsequently, the original experimental data was subtracted from the experimental data of this group to obtain the influence of the Venturi tube cavitation device on the water temperature, as shown in Figure 14 (b). As the inlet pressure gradually increases, both the electrical conductivity and the temperature difference show a gradually increasing trend. This changing trend is consistent with the numerical simulation results, further verifying the accuracy of the simulation results. As illustrated in Figs 14(c) through 14(f), under conditions of fixed Venturi tube length, both conductivity and temperature difference exhibit a progressive decline as the exit cone angle increases incrementally, whereas these parameters demonstrate a corresponding upward trend with increasing throat diameter. When tube length is varied, the conductivity and temperature difference follow a non-linear trajectory: they initially rise and subsequently fall with increasing exit cone angle, while exhibiting a more complex pattern of initial increase, followed by decrease, and final resurgence with increasing throat diameter. Notably, the experimental variation trends of conductivity and temperature difference closely parallel those of the average gas holdup derived from CFD simulations, with comparable magnitudes of fluctuation. This strong correlation implies that changes in conductivity and temperature difference are intrinsically linked to the average gas holdup, thereby providing insight into the dynamic evolution of internal fluid state within the pipeline.

### B. Summary of hydraulic cavitation experiment

Through the hydraulic cavitation experiment, the temperature rise of water and the change of water conductivity were measured to reflect the intensity of cavitation. The changing trends of water temperature difference and water conductivity obtained from the experiment are basically consistent with the simulation results. Although there are some experimental errors, these errors are within the measurement range of the experimental equipment, indirectly verifying the accuracy of the simulation results. However, the water temperature differential and aquatic electrical conductivity merely function as ancillary indicators, failing to fully capture all characteristics of cavitation phenomena. Consequently, future research endeavors should undertake more exhaustive and in-depth experimental investigations to provide more compelling empirical evidence for comprehensive verification.

## VI. Conclusion

Based on FLUENT, this paper studies the hydraulic cavitation effect of Venturi tube, and analyzes the influence of different working conditions and structural parameters on the cavitation effect of Venturi tube through the combination of numerical simulation and experimental verification. Through the above analysis, the following conclusions are drawn:

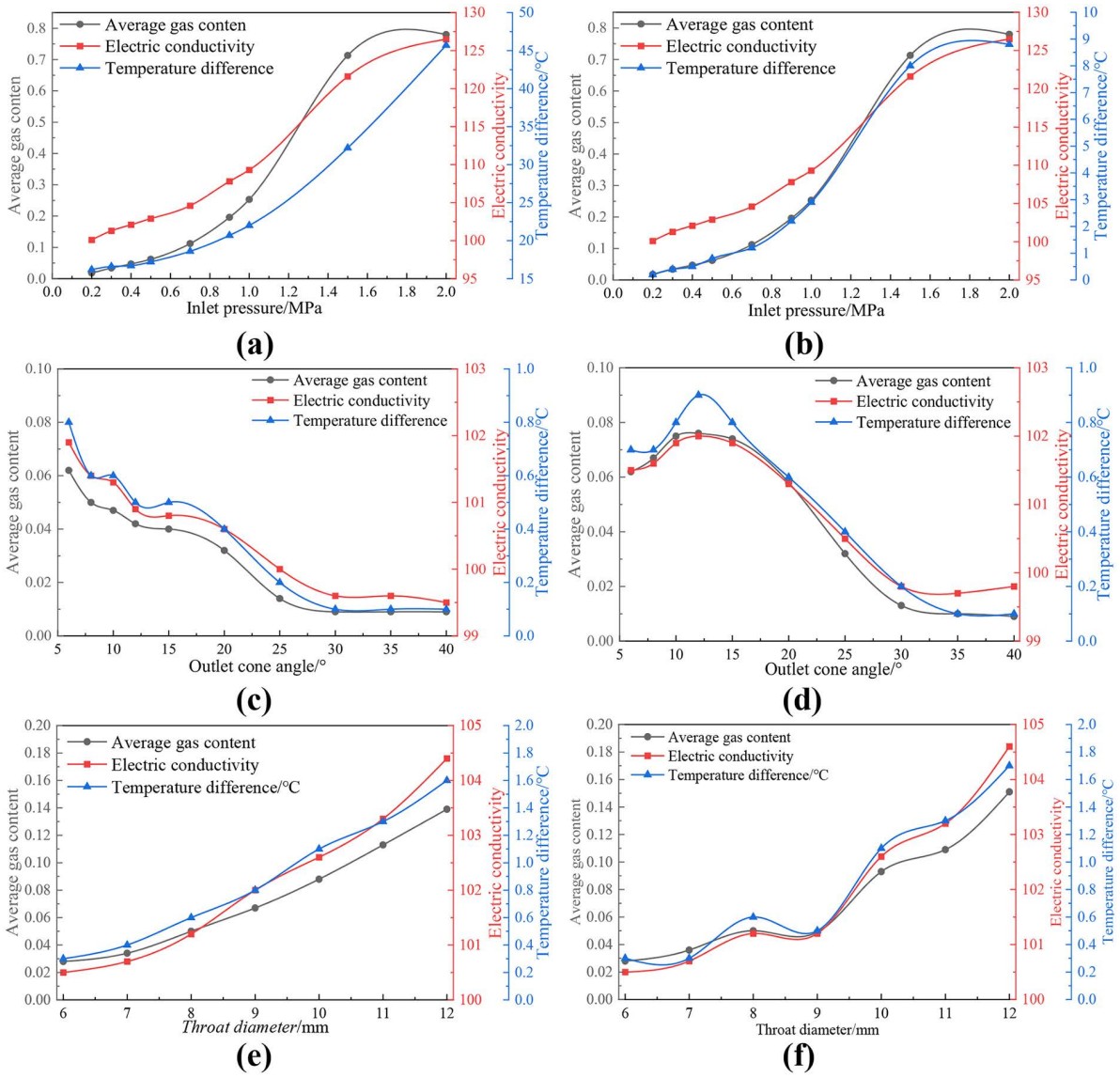

**Fig 14. The changing trends of electrical conductivity and temperature difference in the experiment.** (a-f).

1) With the increase of inlet pressure, the average and maximum gas content of the Venturi tube shows a gradually increasing trend, but the growth rate is gradually slowing down. Therefore, the influence of inlet pressure of the cavitation effect of the Venturi tube is not infinitely positively correlated, but there is a limit value. Under the conditions of this paper, the limit value should be around 1.5 MPa.

2) With the increase of outlet cone angle, the average and maximum gas content of the Venturi tube shows a gradually decreasing trend, and the cavitation effect gradually weakens.

3) With the increase of throat diameter, the average and maximum gas content of the Venturi tube shows a positive effect, and the overall cavitation effect gradually enhances.

4) With the increase of throat length, the average gas content of the Venturi tube is almost not affected, and the maximum gas content gradually decreases, and the overall cavitation effect gradually weakens.

5) The experimental variation patterns of electrical conductivity and temperature difference closely parallel those of the average gas content rate derived from CFD simulations, with comparable magnitudes of fluctuation. This strong correlation implies that variations in electrical conductivity and temperature difference are intrinsically linked to the average gas content rate, thereby providing insight into the dynamic evolution of fluid state within the pipeline.

## Supporting information

**S1 File. Supporting information.**
(RAR)

## Author contributions

**Formal analysis:** Sitong Guo, Xueying Ji, Zhanshan Ma, Xiaolong Zhou.

**Funding acquisition:** Linlin Cao.

**Investigation:** Yunsheng Tian, Zhijie Huang.

**Methodology:** Zhanshuo Zhang, Linlin Cao, Xiaolong Zhou.

**Project administration:** Linlin Cao.

**Supervision:** Xiaolong Zhou.

**Writing – original draft:** Zhanshuo Zhang.

**Writing – review & editing:** Zhanshuo Zhang, Sitong Guo, Xiaobo Liu.

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
