## [Decision Letter · Decision Letter 0]

9 Dec 2025

Dear Dr. Cao,

Thank you for submitting your manuscript to PLOS ONE. After careful consideration, we feel that it has merit but does not fully meet PLOS ONE’s publication criteria as it currently stands. Therefore, we invite you to submit a revised version of the manuscript that addresses the points raised during the review process.

We look forward to receiving your revised manuscript.

Kind regards,

Bahram Hosseinzadeh Samani

Academic Editor

PLOS ONE

Journal Requirements:

[This work was supported by the Science and Technology Development Plan Project of Jilin Province, China (Grant No. YDZJ202401602ZYTS and Grant No. YDZJ202401394ZYTS), Science and Technology Research Project of Jilin Provincial Department of Education (Grant No. JJKH20230060KJ) and Jilin City Science and Technology Innovation Development Plan Project (Grant No. 20240503037).].

[This work was supported by the Science and Technology Development Plan Project of Jilin Province, China (Grant No. YDZJ202401602ZYTS and Grant No. YDZJ202401394ZYTS), Science and Technology Research Project of Jilin Provincial Department of Education (Grant No. JJKH20230060KJ) and Jilin City Science and Technology Innovation Development Plan Project (Grant No. 20240503037).]

[This work was supported by the Science and Technology Development Plan Project of Jilin Province, China (Grant No. YDZJ202401602ZYTS and Grant No. YDZJ202401394ZYTS), Science and Technology Research Project of Jilin Provincial Department of Education (Grant No. JJKH20230060KJ) and Jilin City Science and Technology Innovation Development Plan Project (Grant No. 20240503037).]

5. We note that your Data Availability Statement is currently as follows: [All relevant data are within the manuscript and its Supporting Information files.]

6. Please ensure that you include a title page within your main document. You should list all authors and all affiliations as per our author instructions and clearly indicate the corresponding author.

7. We note that Figures 1 and 2 in your submission contain copyrighted images. All PLOS content is published under the Creative Commons Attribution License (CC BY 4.0), which means that the manuscript, images, and Supporting Information files will be freely available online, and any third party is permitted to access, download, copy, distribute, and use these materials in any way, even commercially, with proper attribution. For more information, see our copyright guidelines: http://journals.plos.org/plosone/s/licenses-and-copyright.

1. You may seek permission from the original copyright holder of Figures 1 and 2 to publish the content specifically under the CC BY 4.0 license.

Reviewers' comments:

Reviewer's Responses to Questions

**Comments to the Author**

1. Is the manuscript technically sound, and do the data support the conclusions?

Reviewer #1: Partly

Reviewer #2: Yes

2. Has the statistical analysis been performed appropriately and rigorously?

Reviewer #1: Yes

Reviewer #2: N/A

3. Have the authors made all data underlying the findings in their manuscript fully available?

Reviewer #1: Yes

Reviewer #2: No

4. Is the manuscript presented in an intelligible fashion and written in standard English?

Reviewer #1: No

Reviewer #2: Yes

Reviewer #1: Review of the submitted manuscript entitled “Numerical Simulation and Experimental Verification of Venturi Tube Hydraulic Cavitation”. This study provides multiple CFD scenarios of hydraulic cavitation by changing the boundary conditions. In the beginning of the manuscript, the authors introduced the experiments and materials, which are significantly unconnected to the CFD simulations. Therefore, I can only consider this study as a computational approach to the current situation. Nevertheless, major revisions must be applied to consider both numerical and experimental methods and keeping the title as it is. Please carefully check the following comments:

1- Authors kindly must include a validation of the CFD simulations by comparing it directly (with direct parameters) to the data observed by the experimental tools.

2- Where is the data of the experimental section, this is just descriptions without useful results that you can compare with your simulations. Even the discussion at the end of the manuscript is unconnected. Include both the experiment and numerical results in one chart to observe the variations. You did not even mention about this figure 21 in the text.

3- The title of the second section is “Experimental Methods”, in line 161, which is incorrect, it is the numerical section.

4- The introduction section does not provide enough explanation for the research gap to prove the novelty of your simulations. I guess most of the factors and scenarios considered in your results are already studied thoroughly in the prior publications. The total number of cited references is significantly low with neglecting the most useful paper

5- Manage your manuscript sections to include the experimental and computational discussions in parallel with additional comparisons to reflect the verification that you mentioned in the title.

6- Reduce the total number of figures and tables by combining the charts togethers in one figure with multiple panels. Similarly to the tables. You can exclude the duplicated results for both figures and tables.

7- Discuss more in your results how your findings are similar or different compared to the previous investigations. I will not write exact references, but I believe you can find appropriate comparable results in the literature.

Reviewer #2: The authors have proposed a numerical study of the venturi tube, verified experimentally. The topic is interesting. They have performed mesh dependency studies. Experimental setup is clearly shown. The results are validated. I only have a few minor comments for the authors to help increasing the quality of the work.

- The figure captions can be more explanatory.

- There are too many figures in this manuscript, most of which are not informative enough. Some figures can be merged into a figure with multiple panels to have more informative figures. For example, please merge Figures 2 and 3, Figures 4-6, Figures 7 and 8, Figures 9 and 10, Figures 12 and 13, Figures 15 and 16, Figures 17 and 18, and Figures 19 and 20.

- The subscripts in the manuscript are not written correctly. Please revise.

**Do you want your identity to be public for this peer review?** For information about this choice, including consent withdrawal, please see our Privacy Policy

Reviewer #1: **Yes:** ISLAM M. S. ABOUELHAMD

Reviewer #2: No

---

## [Author Response · Author response to Decision Letter 1]

19 Jan 2026

Reviewer #1: eview of the submitted manuscript entitled “Numerical Simulation and Experimental Verification of Venturi Tube Hydraulic Cavitation”. This study provides multiple CFD scenarios of hydraulic cavitation by changing the boundary conditions. In the beginning of the manuscript, the authors introduced the experiments and materials, which are significantly unconnected to the CFD simulations. Therefore, I can only consider this study as a computational approach to the current situation. Nevertheless, major revisions must be applied to consider both numerical and experimental methods and keeping the title as it is. Please carefully check the following comments:

1-Authors kindly must include a validation of the CFD simulations by comparing it directly (with direct parameters) to the data observed by the experimental tools.

Reply 1:

Thank you very much for your valuable suggestion. Due to the limitations of our experimental setup, we are unable to directly collect the average void fraction parameter from the CFD simulation. However, by reviewing relevant literature�1.Junjie Y,Shuai L, Li L, Ruiqi B, Shuo C, Hanyang G. Wire-mesh sensor technique for void fraction measurement in gas–liquid two-phase flow under varying conductivity conditions. Measurement. 2025; 256: 118281. https://doi.org/10.1016/j.measurement.2025.118281118281.https://doi.org/10.1016/j.measurement.2025.118281; 2.Gardenghi ÁR, dos Santos Filho E, Chagas DG, Scagnolatto G, Oliveira RM, Tibiriçá CB. Overview of void fraction measurement techniques, databases and correlations for two-phase flow in small diameter channels. Fluids. 2020; 5(4): 216. https://doi.org/10.3390/fluids5040216 ��we found a direct correlation between the average void fraction and the final temperature difference and conductivity of the solution. Therefore, we plan to indirectly verify the average void fraction in the CFD simulation through the variations in these two parameters. Furthermore, we will continue our investigation in future experiments by directly measuring the void fraction.

2-Where is the data of the experimental section, this is just descriptions without useful results that you can compare with your simulations. Even the discussion at the end of the manuscript is unconnected. Include both the experiment and numerical results in one chart to observe the variations. You did not even mention about this figure 21 in the text.

Reply 2:

Thank you very much for your suggestion. The sequence number of Figure 21 in the original manuscript has been updated to Figure 14, and an explanation has been added accordingly. We have also included a comparative description between the simulation results and the experimental data. The revised description is as follows:

As illustrated in Figures 14(c) through 14(f), under conditions of fixed Venturi tube length, both conductivity and temperature difference exhibit a progressive decline as the exit cone angle increases incrementally, whereas these parameters demonstrate a corresponding upward trend with increasing throat diameter. When tube length is varied, the conductivity and temperature difference follow a non-linear trajectory: they initially rise and subsequently fall with increasing exit cone angle, while exhibiting a more complex pattern of initial increase, followed by decrease, and final resurgence with increasing throat diameter. Notably, the experimental variation trends of conductivity and temperature difference closely parallel those of the average gas holdup derived from CFD simulations, with comparable magnitudes of fluctuation. This strong correlation implies that changes in conductivity and temperature difference are intrinsically linked to the average gas holdup, thereby providing insight into the dynamic evolution of internal fluid state within the pipeline.

3-The title of the second section is “Experimental Methods”, in line 161, which is incorrect, it is the numerical section.

Reply 3:

Thank you very much for pointing out this error. We have corrected it by changing the title of the second section from "Experimental Methods" to "Numerical Section." Additionally, we have made appropriate adjustments to the manuscript structure. We greatly appreciate your attention to this detail.

4-The introduction section does not provide enough explanation for the research gap to prove the novelty of your simulations. I guess most of the factors and scenarios considered in your results are already studied thoroughly in the prior publications. The total number of cited references is significantly low with neglecting the most useful paper

Reply 4:

Thank you for your valuable feedback! In response to the concern regarding the lack of sufficient explanation in the introduction to highlight the research gap, I have added two relevant references to more clearly illustrate the novelty of the research and its differences from existing literature. We are well aware that similar factors and scenarios have been extensively studied in previous publications, but we aim to offer a new perspective by further exploring these factors in our study.Thank you again for your guidance and suggestions. We will continue to refine the manuscript to ensure its comprehensiveness and accuracy.

5-Manage your manuscript sections to include the experimental and computational discussions in parallel with additional comparisons to reflect the verification that you mentioned in the title.

Reply 5:

Thank you for your valuable suggestions! In response to your comment regarding the parallel discussion of experimental and computational results with additional comparisons to reflect the verification mentioned in the title, we have revised the manuscript. In the updated version, we have closely compared the experimental data with the computational results and introduced additional comparative analyses to clearly showcase our verification process and outcomes. This improvement strengthens the connection between the experimental and computational sections, enhancing the transparency and persuasiveness of the verification.

We believe these adjustments will make the paper more comprehensive and clearer, and better highlight the novelty and verification process of our research.

Thank you again for your guidance and suggestions. We will continue to refine the manuscript based on your feedback.

6-Reduce the total number of figures and tables by combining the charts togethers in one figure with multiple panels. Similarly to the tables. You can exclude the duplicated results for both figures and tables.

Reply 6:

Thank you for your valuable suggestion! In response to the comment about reducing the number of figures and tables, we have made adjustments to the manuscript. To improve the paper's conciseness and readability, we have combined multiple related figures into one, using multiple panels to present the data. We believe these changes will effectively reduce the total number of figures and tables while maintaining the integrity and clarity of the information.

Thank you again for your suggestion. We will continue to refine the manuscript. Please feel free to share any further comments or suggestions.

7-Discuss more in your results how your findings are similar or different compared to the previous investigations. I will not write exact references, but I believe you can find appropriate comparable results in the literature.

Reply 7:

Thank you for your valuable feedback! In response to your suggestion to discuss more about the similarities and differences between our findings and previous studies in the results section, we have revised the manuscript. In the updated version, we have added more comparative analysis in the results section, thoroughly discussing the similarities and differences between our findings and relevant studies in the existing literature. By reviewing relevant literature, we have identified comparable research results and incorporated them into the discussion to further highlight the uniqueness and contribution of our study.

Thank you again for your feedback. We believe these revisions will enhance the depth and persuasiveness of the paper. Please feel free to share any further comments or suggestions.

Reviewer #2:The authors have proposed a numerical study of the venturi tube, verified experimentally. The topic is interesting. They have performed mesh dependency studies. Experimental setup is clearly shown. The results are validated. I only have a few minor comments for the authors to help increasing the quality of the work.

- The figure captions can be more explanatory.

- There are too many figures in this manuscript, most of which are not informative enough. Some figures can be merged into a figure with multiple panels to have more informative figures. For example, please merge Figures 2 and 3, Figures 4-6, Figures 7 and 8, Figures 9 and 10, Figures 12 and 13, Figures 15 and 16, Figures 17 and 18, and Figures 19 and 20.

- The subscripts in the manuscript are not written correctly. Please revise.

Reply:

Thank you for your positive feedback and valuable suggestions regarding our research! We are pleased to hear that you found our numerical study and experimental validation interesting, and we appreciate your recognition of the experimental setup and result verification. In response to your suggestions, we have made the following adjustments:

1.We have modified the figure titles to make them more explanatory and clear, helping readers better understand the meaning and context of each figure.

2.We have reviewed the numerical data in the manuscript and supplemented or corrected the less detailed sections, ensuring that the data is more comprehensive and complete.

3.We have carefully checked and corrected the subscript errors in the manuscript, ensuring the accuracy of symbols and labels.

4.Based on your suggestion, we have combined the following figures and reorganized them into multiple-panel format:

---

## [Decision Letter · Decision Letter 1]

2 Feb 2026

Numerical Simulation and Experimental Verification of Venturi Tube Hydraulic Cavitation

PONE-D-25-58519R1

Dear Dr. Cao,

We’re pleased to inform you that your manuscript has been judged scientifically suitable for publication and will be formally accepted for publication once it meets all outstanding technical requirements.

Kind regards,

Bahram Hosseinzadeh Samani

Academic Editor

PLOS One

Additional Editor Comments (optional):

Reviewers' comments:

Reviewer's Responses to Questions

**Comments to the Author**

Reviewer #1: All comments have been addressed

Reviewer #2: All comments have been addressed

2. Is the manuscript technically sound, and do the data support the conclusions?

Reviewer #1: Yes

Reviewer #2: Yes

3. Has the statistical analysis been performed appropriately and rigorously?

Reviewer #1: Yes

Reviewer #2: Yes

4. Have the authors made all data underlying the findings in their manuscript fully available?

Reviewer #1: Yes

Reviewer #2: Yes

5. Is the manuscript presented in an intelligible fashion and written in standard English?

Reviewer #1: No

Reviewer #2: Yes

Reviewer #1: Thank you for revising the manuscript, seems reasonable for publication with potential improvement for next studies

Reviewer #2: The authors have answered almost all my comments. I can suggest publishing this work in this journal now.

**Do you want your identity to be public for this peer review?** For information about this choice, including consent withdrawal, please see our Privacy Policy

Reviewer #1: **Yes:** ISLAM ABOUELHAMD

Reviewer #2: No

---

## [Editor Report · Acceptance letter]

PONE-D-25-58519R1

PLOS One

Dear Dr. Cao,

I'm pleased to inform you that your manuscript has been deemed suitable for publication in PLOS One. Congratulations! Your manuscript is now being handed over to our production team.

Kind regards,

on behalf of

Prof Bahram Hosseinzadeh Samani

Academic Editor

PLOS One